# miRNA Clusters with Down-Regulated Expression in Human Colorectal Cancer and Their Regulation

**DOI:** 10.3390/ijms21134633

**Published:** 2020-06-29

**Authors:** Paulína Pidíkova, Richard Reis, Iveta Herichova

**Affiliations:** 1Department of Animal Physiology and Ethology, Faculty of Natural Sciences, Comenius University in Bratislava, 842 15 Bratislava, Slovakia; paulina.pidikova@uniba.sk; 2First Surgery Department, University Hospital, Comenius University in Bratislava, 811 07 Bratislava, Slovakia; reis.richard.477@gmail.com

**Keywords:** proliferation, apoptosis, chemoresistance, survival, long ncRNA, methylation, angiogenesis, cell adhesion

## Abstract

Regulation of microRNA (miRNA) expression has been extensively studied with respect to colorectal cancer (CRC), since CRC is one of the leading causes of cancer mortality worldwide. Transcriptional control of miRNAs creating clusters can be, to some extent, estimated from cluster position on a chromosome. Levels of miRNAs are also controlled by miRNAs “sponging” by long non-coding RNAs (ncRNAs). Both types of miRNA regulation strongly influence their function. We focused on clusters of miRNAs found to be down-regulated in CRC, containing miR-1, let-7, miR-15, miR-16, miR-99, miR-100, miR-125, miR-133, miR-143, miR-145, miR-192, miR-194, miR-195, miR-206, miR-215, miR-302, miR-367 and miR-497 and analysed their genome position, regulation and functions. Only evidence provided with the use of CRC *in vivo* and/or *in vitro* models was taken into consideration. Comprehensive research revealed that down-regulated miRNA clusters in CRC are mostly located in a gene intron and, in a majority of cases, miRNA clusters possess cluster-specific transcriptional regulation. For all selected clusters, regulation mediated by long ncRNA was experimentally demonstrated in CRC, at least in one cluster member. Oncostatic functions were predominantly linked with the reviewed miRNAs, and their high expression was usually associated with better survival. These findings implicate the potential of down-regulated clusters in CRC to become promising multi-targets for therapeutic manipulation.

## 1. Introduction

Colorectal cancer (CRC) is the fourth-most common cancer worldwide with high mortality [1]. In spite of progress in CRC diagnostics and the determination of patient prognosis, there is still a need for improvement. During last two decades, miRNAs have been frequently discussed as a potential tool for the assessment of cancer progression [2].

MicroRNAs (miRNAs) belong to a large family of non-coding RNAs (ncRNAs). The average length of miRNAs is only 22 nucleotides (nt). The canonical pathway of miRNA synthesis begins with transcription from a DNA template, similarly to mRNA, creating primary-miRNAs (pri-miRNAs). After transcription that is usually mediated by RNA polymerase II, pri-miRNAs are capped by a 5′-7-methyl guanosine cap and polyadenylated [3]. Pri-miRNAs are characterised by a hairpin and flanking overhangs of single-stranded RNA. This structure is recognised by the microprocessor complex, which is composed of RNase III Drosha and DiGeorge syndrome critical region gene 8 protein (DGCR8) dimer, which cleaves pri-miRNA at the stem of the hairpin to produce pre-miRNA from the precursor [3].

This structural change of pre-miRNA hairpin allows for the interaction of pre-miRNAs with the nuclear transport receptor Exportin 5 and Ran GTPase, which facilitate the translocation of pre-miRNAs from the nucleus into the cytoplasm. Next, RNase III Dicer is recruited to cut out a single-stranded loop. In this way, Dicer produces a mature miRNA duplex with two to three nt overhangs at both ends. The mature miRNAs sequence excised from the 5′ arm is designated as 5p, and the sequence excised from the 3′ arm is designated as 3p [4,5].

Two strands of mature miRNAs have different roles in miRNA signalling executed by the RNA-induced silencing complex (RISC). To assemble the RISC, cooperation between Dicer, mature miRNAs, some member of the Argonaute protein family (Ago) and transactivation response element RNA-binding protein (TRBP) are needed. After loading of the miRNA into the RISC and rewinding, one miRNA strand, the so-called guide strand is (via Ago) positioned in a conformation that allows for target mRNA pairing [6]. mRNA recognition is based on complementarity between the miRNA response element (MRE), which is frequently located in the 3′ untranslated region (3′ UTR) of an mRNA, and the “seed” sequence of miRNA, which is a segment between 2 and 8 nt from the 5′ end. miRNAs exert an inhibitory influence that is dependent on homology between miRNAs and target mRNA. If complementarity is high, the target mRNA is degraded, whereas if homology is less extensive, translation repression occurs [7]. Particular MRE is usually present at several mRNAs that are all targeted by corresponding miRNAs [8].

The second miRNAs strand, the so-called passenger strand, is present in the cytoplasm at much lower concentration compared with the guide strand and, in spite of its low concentration, can be to some extent also incorporated into the RISC [4,8,9].

miRNAs that are closely located in the genome create miRNA clusters. It has been shown that 20–40% of more than 1800 human miRNA sequences are organised in polycistrons (clusters) that are transcribed together [5,10,11]. miRNA clusters are usually composed of 2 to 8 members, but more than 60% of clusters contain only two miRNAs [12]. miRNA clusters in the human genome are divided into two groups—homologous clusters and heterologous clusters. Homologous clusters are composed of miRNAs from the same family [5]. Gene families are groups of homologous genes that are likely to have highly similar functions mainly because of the same seed sequence [5]. miRNA families and miRNA clusters have complex distributions in the genome. One miRNA family may be located in one or more clusters, and one cluster may be involved in one or more families. More than 60% of miRNA clusters in the human genome contain miRNAs from the same family [12].

The abovementioned miRNA organisation is also mirrored in their names. miRNAs with identical mature sequences but different precursor hairpins and locations in the genome are designated with a number in the suffix [13]. For example, miR-133a-1 is located on chromosome 18 of the human genome and miR-133a-2 is located in chromosome 20, but both precursors are processed into a final mature miRNA with the same sequence [14]. miRNAs that differ only in one or two positions in their sequence can be distinguished by letter suffixes in the name (e.g., miR-133a and miR-133b) [13].

Long ncRNAs (lncRNAs) are typically 1000–10,000 nt long and, according their structure, can be split into linear lncRNAs (including pseudogenes) and circular RNAs (circRNAs) [15]. Similarly to miRNA, lncRNAs are transcribed from exonic, intronic or intergenic DNA sequences frequently showing polycistronic organisation; however, they utilise much more genome information compared to miRNAs [16]. Transcription of linear lncRNAs shows high similarity to that described for protein-coding genes, except for the presence of the translated open-reading frame. lncRNAs strongly influence miRNA expression [7].

## 2. Biogenesis of miRNA

In spite of huge progress in the measurement of miRNAs [2] there remain inconclusive and contradictory results about miRNA up- or down-regulation in CRC, e.g., miR-204, miR-203, miR-200, miR-150, miR-142 [17]. This inconsistency precludes these miRNAs from use as effective bio-tools. A recent review suggested that miRNAs organised in clusters may be more reliable biomarkers as they can share the same way of transcriptional regulation. We focused on down-regulated clusters, as their levels are less likely to be masked by cell fragmentation due to cell death.

Regulation of miRNA expression is frequently associated with their localisation in the genome. Approximately 40–60% of miRNAs are located in intronic areas of protein-coding genes or nonprotein-coding transcripts [5,7,18,19]. Many intronic miRNAs are expressed together with their host genes in one polycistronic transcript, and it is likely that their expression is regulated by a promoter of the host gene [11,18,20]. On the other hand, approximately one-third of intronic miRNAs in the human genome have their own promoters and may be transcribed independently of their host gene promoter [21,22]. Methylation of the promoter of miRNAs or their host gene promoters also contributes to regulation of miRNA expression (e.g., [23,24,25]).

Promoter-independent regulation of miRNAs is executed by competing endogenous RNAs (ceRNAs) that interact with miRNAs. Both types, linear as well as circular lncRNAs, can inhibit miRNA function by binding based on complementary sequences and prevent the interaction of miRNAs with target mRNAs. This mechanism is known as “sponging” [26]. Since miRNAs have been previously shown to play an important role in cancer progression [8], the effects of ceRNAs as modulators of miRNA activity are also of crucial importance in this respect [15].

Finally, it has been shown that miRNAs show sex-dependent regulation of expression. By comparison of the miRNA transcriptomes of males and females, it was revealed that there are 73 female-biased and 163 male-biased miRNAs in the human circulation and tissues [27]. A difference in miRNA expression was also observed in colorectal cancer (CRC) tissue [9,28,29]. A reason for this finding has not been completely elucidated; however, it does not seem to be associated with the location of miRNAs on sex chromosomes [27]. The role of steroid hormones has been investigated in this respect [25]. A network of oestrogen-responsive miRNAs has been implicated in the development of sex-dependent features [30]. It has also been shown that oestrogen regulates miRNA expression in many stable cancer cell lines [31,32]. Other mechanisms of miRNA regulation are extensively reviewed elsewhere [25].

Since CRC is one of the leading causes of cancer mortality worldwide, recent review has been focused on the regulation of miRNA clusters formed from miRNAs that are deregulated in this disease. We focused on clusters with decreased expression because a convincing majority of studies using CRC tissues or corresponding models have shown the oncostatic capacity of these clusters and support their therapeutic potential.

## 3. miRNA Clusters Down-Regulated in Human CRC

Only miRNA genes in the same orientation, and not separated by a transcription unit or a miRNA in the opposite orientation, located within 50 kb of distance were recognised as clusters [20]. 

In the following section, we analyse the available information about their regulation by transcription factors, lncRNAs and methylation, tumour suppressor or oncogenic potential and target genes in CRC.

### 3.1. Clusters miR-100/let-7a-2/miR-125b-1, miR-99a/let-7c and miR-99b/let-7e/miR-125a

Clusters miR-99a/let7c, miR-99b/let-7e/miR-125a and miR-100/let-7a/miR-125b-1 are located on separate chromosomes (Table 1), but are functionally related as they are composed of members belonging to the same families (Appendix A). Known information about the presence of miRNA-specific transcription start sites (TSSs) is shown in Table 1. 

Information about clusters miR-99a/let7c, miR-99b/let-7e/miR-125a and miR-100/let-7a/miR-125b-1 regulation via transcription factors and lncRNAs is not always available for all clustered miRNAs in CRC models. The doublecortin-like kinase 1 (DCLK1) [33] and lncRNA *ANRIL* [34] have been found to be negative regulators of let-7a-5p expression in CRC cell lines. Expression of miR-125a-5p is strongly influenced by hypermethylation in CRC tumours [23], and it has been shown to be sequestered by lncRNA *HOXA11-AS* [35] and circRNA *VAPA* (Appendix A) [36]. miR-125b-5p/3p is subjected to complex regulation, including via the transcription factors peroxisome-proliferator-activated receptor gamma (PPARG), nuclear factor kappa B subunit 1 (NFKB1), tumour protein p53 (p53), MYC proto-oncogene, bHLH transcription factor (MYC), caudal type homeobox 2 (CDX2), lncRNA NEAT1, MEG3, UCA1 and MALAT1 [37], and by methylation [23].

Levels of let-7c,-5p let-7e-5p, miR-99a-5p, miR-100-5p, miR-125a-5p and miR-125b-5p have been found to be decreased in CRC tumours compared to adjacent tissue (Appendix A). There is insufficient information to make statements about the deregulation of miR-99b in CRC. Most of the miRNAs belonging to the abovementioned clusters are positively associated with better survival and show tumour-suppressive functions (Appendix A), which are executed via a wide range of target genes (Appendix A).

let-7a-5p induces cell cycle arrest and reduces cell growth through targeting genes encoding ubiquitin like with PHD and ring finger domains 2 (*UHRF2*) [38], the Rho effector rhotekin (*RTKN*) [39] and MYC [33] in CRC cell lines. The known target of let-7a-3p is the ABC transporter ATP-binding cassette subfamily C member 1 (*ABCC1*), which is involved in the development of cell chemoresistance [34].

Low expression of let-7c-5p is associated with metastasis and cell growth in CRC tissues and up-regulation of let-7c-5p in the highly metastatic Lovo cell line caused a decrease in migration and inhibition of cell growth through targeting matrix metallopeptidase 11 (*MMP11*) and PBX homeobox 3 gene (*PBX3*) [40].

Increased expression of let-7e-5p in CRC cell lines leads to decreased cell migration and proliferation through targeting the gene coding for serine/threonine kinase *DCLK1* [41], increased sensitivity to treatment with 5-fluorouracil (5FU) and decreased cell invasion through targeting ST8 alpha-N-acetyl-neuraminide alpha-2,8-sialyltransferase 1 (*ST8SIA1*) [42]. let-7e-5p also induces cell cycle arrest through targeting genes encoding insulin-like growth factor 1 receptor (*IGF1R*), which also mediates the decreased sensitivity of CRC cells to both radio- and chemotherapy [43,44].

The miR-99a-5p [65] and miR-99b-5p [66] target gene coding serine/threonine protein kinase with oncogenic potential is called mechanistic target of rapamycin kinase (*MTOR*) in CRC cell lines. 

miR-100-5p inhibits cell growth, induces apoptosis and decreases cell invasion, possibly via targeting RAP1B, a member of RAS oncogene family (*RAP1B*) [67].

miR-125a-5p induces apoptosis through targeting the BCL2 apoptosis regulator (*BCL2*) and BCL2 family members BCL2-like 12 (*BCL2L12*) and myeloid cell leukaemia 1 gene (*MCL1*) [68] in CRC cell lines. miR-125a-5p also executes its functions via targeting genes coding for peptidyl arginine deiminase 2 (*PADI2*) involved in the promotion of metastasis [35] and pro-angiogenic vascular endothelial growth factor A (*VEGFA*) [69]. miR-125a-5p also inhibits cell proliferation and migration by targeting SMAD-specific E3 ubiquitin protein ligase 1 (*SMURF1*) [70], phospholipid:diacylglycerol acyltransferase, called tafazzin (*TAZ*) [71] and cAMP-responsive element-binding protein 5 (*CREB5*) [36]. Overexpression of miR-125a-3p inhibits cell proliferation and migration through targeting fucosyltransferases 5 and 6 (*FUT5* and *FUT6*, respectively) [72].

Overexpression of miR-125b-5p leads to promotion of apoptosis and blockage of cell cycle progression in the human CRC cell line HCT-8. On the other hand, HCT-8 cells with high expression of miR-125b show more invasive and metastatic potential through the promotion of epithelial–mesenchymal transition (EMT). One of the validated targets of miR-125b-5p is the anti-apoptotic gene *MCL1* [73] for which high expression is associated with shorter survival times in CRC patients [74]. Another target gene of miR-125b-5p is the APC regulator of the WNT signalling pathway (*APC*) gene. Negative correlation between expression of miR-125b-5p and *APC* has also been confirmed in tumour tissue from patients with CRC [75].

### 3.2. Clusters miR-1-2/133a-1, miR-1-1/133a-2 and miR-206/133b

The miR-1-1/133a-2, miR-1-2/133a-1 and miR-206/133b clusters are encoded on different chromosomes (Table 1). miR-206-3p differs from miR-1-3p only by four nt [14]. Members of the abovementioned clusters are traditionally considered to be muscle-specific miRNAs, but growing evidence supports a broader expression pattern, including CRC tissues [14].

miR-1 and miR-133a expression is silenced by DNA hypermethylation [76]. Expression of miR-206-3p is induced by C–C motif chemokine ligand 19 (CCL19) [77]. In addition, miR-133a-3p in CRC cells is sponged by the complementary lncRNAs *ABHD11-AS1* and *XIST* [78,79], while miR-133b-3p has been shown to be sequestered by the lncRNAs *LINC00114* [80], *ENSG00000231881* [81] and *LINC00467* [82]. miR-206-3p is a target of the lncRNA *LINC00707* [83,84].

The expression of miR-1-3p, miR-133a-3p, miR-133b-3p and miR-206-3p is decreased in CRC tissue compared to normal tissue, and high levels are associated with better survival in patients with CRC (Appendix A). Because of the inhibitory influence of miR-1-3p, miR-133a-3p, miR-133b-3p and miR-206-3p on cell growth, migration, proliferation and chemoresistance, they are considered to be tumour suppressors (Appendix A), executing their roles via inhibition of a wide range of target genes (Appendix A). 

miR-1-3p expression shows negative correlation with tumour size, degree of differentiation, lymph node metastasis and tumour/nodus/metastasis stage (TNM) [85,86,87]. In vitro, miR-1-3p suppresses cell growth, migration, motility and glycolysis by targeting *VEGF* [87] and notch receptor 3 (*NOTCH3*), which is known to be crucially involved in developmental processes by controlling cell fate decisions [88] as well as hypoxia-inducible factor 1 subunit alpha (*HIF1A*) [89]. The target genes of miR-1-3p also encode for NLR family apoptosis inhibitory protein (*NAIP*), which plays a role in the inhibition of apoptosis [86], and the focal adhesion protein LIM and SH3 protein 1 (*LASP1*) [85].

miR-206-3p suppresses CRC cell migration [90], cell proliferation and accelerates apoptosis via targeting the oncogene formin-like 2 (*FMNL2*), MET proto-oncogene, receptor tyrosine kinase (*MET*) [91], *NOTCH3* [92], tetraspanin-like protein called transmembrane 4 L six family member 1 (*TM4SF1*) [93] and *BCL2* [94]. These oncostatic effects are prevented by the overexpression of lncRNA *LINC00707* [83,84].

miR-133a-3p inhibits both in vitro and in vivo cell growth via the inhibition of ring finger and FYVE-like domain-containing E3 ubiquitin protein ligase (*RFFL*), which induces degradation of p53 protein [95], *LASP1* [96], fascin actin-bundling protein 1 (*FSCN1*) (involved in the regulation of cell motility) [97], oncogenic SUMO-specific peptidase 1 (*SENP1*) [98] and an RNA helicase called eukaryotic translation initiation factor 4A1 (*EIF4A1*) [99].

miR-133b-3p inhibits cell invasion and induces apoptosis; these effects are reversed by overexpression of its target gene C-X-C motif chemokine receptor 4 (*CXCR4*) [100]. miR-133b-3p targets epidermal growth factor receptor (*EGFR*) and shows synergistic oncostatic effects with cetuximab [101]. The oncostatic effects of miR-133b are also executed by inhibition of homeobox A9 (*HOXA9*) and metastasis inducer zinc finger E-box binding homeobox 1 (*ZEB1*) [102]. miR-133b-3p targets *MET* (involved in invasive tumour growth) [103], the gene encoding ferritin light chain (*FTL*), lncRNA *LINC00467* (promoting CRC cell resistance against 5FU) [82], a component of the nuclear pore complex proto-oncogene nucleoporin 214 (*NUP214*) [45] and some others (Appendix A).

### 3.3. Clusters miR-192/194-2 and miR-215/194-1

Clusters miR-192/194-2 and miR-215/194-1 are located on different chromosomes (Table 1) and consist of miR-194 and miR-192 or miR-215, which differ by only two nt [56].

Expression of clusters miR-192/194-2 and miR-215/194-1 is induced by p53 in the human colon cancer cell line HCT116 [57,104], and miR-194-5p expression is stimulated by a hepatocyte nuclear factor called HNF1 homeobox A (HNF1A) via binding to the miR-194 promoter [56]. An inhibitor of miR-194-5p expression is the non-histone chromosomal protein called high mobility group AT-hook 2 (HMGA2), which exerts its function via upstream promoters of both miR-194 loci [105]. Expression of miR-215-5p in CRC cells is regulated by caudal-type homeobox 1 (CDX1) independently of other members of the miR-194 cluster [59]. miR-194-5p is also regulated by sponging with lncRNA *H19* [106], and the opposite strand, miR-194-3p, is sequestered with lncRNA *TP73-AS1* in CRC cell lines [107]. miR-215-5p is sponged by lncRNAs *UICLM* [108] and *FTX* [109]. Expression of miR-215-5p increases under hypoxic conditions [110] and after melatonin treatment [111] in CRC cell lines.

Expression of miR-192-5p, -194-5p and -215-5p has been shown to be down-regulated in colon cancer tissue compared to normal tissue (Appendix A). While expression of miR-192 and -194 is associated with better survival in patients with CRC, the association of miR-215 expression with better survival is not conclusive yet (Appendix A).

Most of the reports about the functions of miR-192/194-2 and miR-215/194-1 clusters indicate their tumour-suppressive roles (Appendix A), as cell cycle arrest and inhibition of cell adhesion are observed after their overexpression. These functions are usually executed via the silencing of their target genes (Appendix A).

miR-194-5p targets several genes involved in regulation of cell growth. Inhibition of mitogen-activated protein kinase kinase kinase 4 (*MAP4K4*) by a miR-194-5p mimic causes a decrease in cell proliferation under in vivo and in vitro conditions [112]. Overexpression of another target gene of miR-194-5p transcriptional activator called forkhead box M1 (*FOXM1*) reversed the effects of the miR-194-5p mimic under in vitro conditions [106]. miR-194-5p is also involved in regulation of the Wnt/β-catenin pathway through targeting AKT serine/threonine kinase 2 (*AKT2*), which contributes to the activation of Wnt/β-catenin signalling [113]. Another target gene of miR-194-5p is an endoplasmic reticulum contact protein called VAMP associated protein A (*VAPA*), which contributes to the regulation of vesicular transport with a positive effect on cell survival [105]. The diversity of miR-194-5p functions has been pointed out after it was found that miR-194-5p also targets a negative regulator of angiogenesis thrombospondin 1 (*THBS1*) and promotes angiogenesis [104]. Another target gene of miR-194-3p is transforming growth factor alpha (TGFA), which has an oncogenic role in CRC [107].

miR-192-5p decreased the liver metastasis of colon cancer in an orthotopic mouse model of colon cancer through targeting the expression of several oncogenic genes, including anti-apoptotic *BCL2*, Wnt/β-catenin activator called zinc finger E-box binding homeobox 2 (*ZEB2*) and pro-angiogenic *VEGFA* [114].

Overexpression of miR-215-5p in CRC cells leads to decreased migration and proliferation through targeting the transcription factor *YY1* [115]. Cell proliferation is suppressed by miR-215-5p through targeting the G2/M checkpoint regulator called denticleless E3 ubiquitin protein ligase homolog (*DTL*) [116,117]. Clonogenicity inhibition mediated by miR-215-5p is exerted by targeting the epidermal growth factor family member epiregulin (*EREG*) and transcriptional inducer homeobox B9 (*HOXB9*) [118]. miR-215-5p in CRC cells induces differentiation through targeting BMI1 proto-oncogene, polycomb ring finger (*BMI1*) [59]. miR-215-5p in CRC also targets Wnt/β-catenin activator *ZEB2*, which is involved in the regulation of EMT [108,109]. As there is a high degree of homology between miR-215-5p and miR-192-5p, they both target the mediator of angiogenesis called sushi repeat-containing protein X-linked 2 (*SRPX2*) [119]. Chemoresistance to 5FU is influenced by miR-215-5p via targeting thymidylate synthetase (*TYMS*), which catalyses the dTMP biosynthesis necessary for DNA synthesis [111]. Resistance to chemotherapy is also regulated by the passenger strand miR-215-3p, which increases sensitivity to 5FU by targeting C-X-C motif chemokine receptor 1 (*CXCR1*) [120].

### 3.4. Clusters miR-15a/16-1 and miR-15b/16-2

miR-15/16 is present in the human genome in the form of two paralogues, miR-15a/16-1 and miR-15b/16-2 (Table 1). Expression of miR-15 and miR-16 is regulated by their host gene promoter (Table 1). Moreover, miR-15a-5p is sponged by lncRNA *LINC00473* in CRC cell lines [121]. Expression of miR-15b-5p is inhibited by sirtuin 1 (SIRT1), which prevents transcriptional activator AP-1 from binding to the miR-15b-5p promoter [122]. miR-16-5p is sponged by lncRNA *SNHG12* in several CRC cell lines [123].

Although a decrease in miR-15-5p/16-5p in CRC tissue compared to normal tissue has been reported more frequently than the opposite, there are also studies implicating the up-regulation of miR-15/16 expression (Appendix A). Similarly, better survival is more frequently linked to high expression of miR-15/16 members; however, a worse survival association with high miR-15/16 expression has also been documented (Appendix A).

Generally, tumour-suppressive functions have been attributed to miR-15/16 clusters. Increased expression of miR-15a-5p and miR-16-1-5p reduced tumour growth in the colons of nude mice [124], and higher expression of miR-15a-5p led to suppressed proliferation of colon cancer cells in vitro [125]. These effects are mostly mediated by miR-15/16 target genes (Appendix A).

Common targets of miR-15a-5p and miR-16-5p in CRC cell lines are cyclin B1 (*CCNB1*) [124] and transcription factor AP-4 (*TFAP4*), which is involved in the regulation of EMT [126]. miR-15a-5p inhibits cell growth by targeting pro-survival protein *BCL2* [125,127], a regulator of stemness called SRY-box transcription factor 2 (*SOX2*) [125], the oncogene Yes1 associated transcriptional regulator (*YAP1*), *DCLK1* and *BMI1*, which facilitates cell invasion and migration [127].

Overexpression of miR-16-5p in CRC cell lines decreases cell migration and proliferation through targeting KRAS proto-oncogene, GTPase (*KRAS*) both in vivo and in vitro [128]. miR-16-5p levels negatively correlate with expression of a VEGF receptor called kinase insert domain receptor (*KDR*) and the MYB proto-oncogene, transcription factor (*MYB*) [129]. miR-16-5p is involved in the induction of apoptosis and cell growth inhibition by targeting integrin subunit alpha 2 (*ITGA2*) [130] and survivin (*BIRC*) [131]. The target genes of miR-16-5p in CRC cells are also *CDX2* (active mainly during development) [132] and prostaglandin-endoperoxide synthase 2 (*PTGS2*), which catalyses the first step in the synthesis of prostanoids [133].

miR-15b-5p decreases cell proliferation through targeting growth via the Pim-1 proto-oncogene, serine/threonine kinase (*PIM*) in CRC cell lines [134]. miR-15b-5p increases sensitivity to chemo- and radiotherapy by targeting *DCLK1* [135], *NFKB1* and a kinase called component of inhibitor of nuclear factor kappa B kinase complex (*CHUK*) [136]. *NFKB1* and *CHUK* are both associated with the NF-κB pathway [136]. miR-15b-5p also decreases cell migration through targeting of the first enzyme in the fatty acid oxidation pathway acyl-coenzyme A oxidase 1 (*ACOX1*) [122]. On the other hand, overexpression of miR-15b-5p in CRC cell lines increases colony formation by targeting the tumour suppressor klotho (*KL*) and MTSS I-BAR domain-containing 1 (*MTSS1*) [137].

### 3.5. Cluster miR-143/145

Bicistronic miR-143/145 is negatively regulated by Ras-responsive element-binding protein 1 (RREB1) [138], and the expression of both miRNAs is suppressed via EGFR [139]. The core promoter region of miR-145 is regulated by histone methylation in CRC cell lines [62] and snail family transcriptional repressor 1 (SNAI1) [140]. miR-145-5p is sponged by circRNA *CIRC_001569* [141], snoRNA *SNHG1* [142], lncRNA *SOX21-AS1* [143], lncRNA *CASC15* [144], circRNA *PVT1* [145] and lincRNA-*ROR* [146]. miR-143-3p is sequestered by lincRNA *UCC* [147] and ceRNA *PART-1* in SW620 cells [148].

Expression of miR-143-3p and miR-145-5p is significantly decreased in CRC tissue compared to normal tissue, and in both cases, decreased expression was associated with shorter survival time and increased disease recurrence (Appendix A).

Expression of miR-143-3p/145-5p is negatively associated with CRC clinicopathological features and exerted oncostatic effects, mainly via influencing their target genes (Appendix A).

The miR-143/145 cluster is involved in the regulation of several key components of the KRAS signalling pathway [138,139]. miR-145-5p is involved in the inhibition of cell proliferation and migration via targeting genes encoding *NAIP* [86], fascin actin-bundling protein 1 (*FSCN1*) involved in regulation of cell motility [149], the focal adhesion protein paxillin (*PXN*) that facilitates cellular contact with the underlying extracellular matrix [150] and *ZEB2* [151]. ETS transcription factor ERG (*ERG*), which is up-regulated in CRC tumours (however, its role in this tissue is not completely elucidated) [152], and E2F transcription factor 5 (*E2F5*), which is involved in cell cycle control [141], are also targeted by miR-145-5p. miR-145-5p influences cancer invasiveness via the inhibition of BAG cochaperone 4 (*BAG4*) and formin-like 2 protein (*FMNL2*) [141]. The cell cycle is influenced by miR-145-5p by targeting G1 regulators cyclin-dependent kinase 6 (*CDK6*), cyclin D2 (*CCND2*), E2F transcription factor 3 (*E2F3*) and *MYC* [139]. miR-145-5p inhibits the metastatic CRC cell invasion induced by *LASP1* [62] and targets myosin VI (*MYO6*), which promotes cell growth in the SW1116 cell line [143].

Among the target genes of miR-143-3p are hexokinase 2 (*HK2*), which causes a decrease in lactate production after inhibition mediated by miR-143-3p [153], toll-like receptor 2 (*TLR2*) [154] and catenin delta 1 (*CTNND1*) [155], which are involved in regulation of cell invasion and migration. miR-143-3p also targets *PTGS2*, *KRAS* and a member of the MAPK family mitogen-activated protein kinase 7 (*MAPK7*) [139], integrin subunit alpha 6 (*ITGA6*) and ArfGAP with SH3 domain, ankyrin repeat and PH domain 3 (*ASAP3*), with roles in the development of metastasis [156]. In addition to cell migration, tumour growth and angiogenesis in CRC inhibition in vivo and in vitro, miR-143-5p contributes to an increase in chemosensitivity of CRC cells to oxaliplatin via targeting *IGF1R* [157].

### 3.6. Cluster miR-302b/302c/302a/302d/367

This polycistron codes for miRNAs with high homology sequences, showing differences only in the last six nt on the 3´end [63,158]. Expression of miR-302c-3p has been shown to be regulated by methylation [24] and is sponged by lncRNA *SNHG16* [159].

Expression of miR-302a-3p and -302c-3p is decreased in CRC tissue compared to normal tissue, and high expression of miR-302a and -302c is associated with better survival (Appendix A).

miR-302a-3p up-regulation suppresses the growth and invasion of SW480 and HCT116 cells, accompanied by a reduction in the expression of matrix metallopeptidase 9 and 2 (*MMP9* and *MMP2*, respectively). The inhibitory effects of miR-302a-3p are mediated via the MAPK and PI3K/Akt signalling pathways [160]. The tumour suppressor role of miR-302a-3p is also executed by targeting nuclear factor IB (*NFIB*) and the induction of cetuximab chemosensitivity, which is caused by suppressing cell-surface expression of the glycoprotein CD44 [161]. miR-302a-3p also induces 5FU sensitivity and viability inhibition via the inhibition of *IGF1R* [162]. Expression of mir-302a-3p is decreased in human CRC cell lines after the induction of autophagy by treatment with 5FU or starvation [163]. 

miR-302c-3p levels negatively correlate with lymph node metastases, tumour invasion and advanced TNM stage [24]. Overexpression of miR-302c-3p in CRC cells causes a decrease in cell growth and stimulates apoptosis [24,164]. Overexpression of miR-302c-3p promotes sensitivity in CRC cell lines to 5FU and oxaliplatin via targeting PLAG1 zinc finger (*PLAG1*), with oncogenic potential [24], and the ABC transporter called ATP-binding cassette subfamily B member 1(*ABCB1*) [165], respectively. Another target gene of miR-302c-3p is transcription factor AP-4 (*TFAP4*), which is involved in the promotion of EMT and cell migration [164] (Appendix A).

### 3.7. Cluster miR-497/195

Cluster miR-497/195 does not have a known paralogue (Table 1). The functions of miR-497-5p are influenced by sponging with lncRNAs *SNHG1* [166], *TTN-AS1* [167] and *AC009022.1* [168] and via methylation of its promoter [64]. miR-195-5p levels are regulated by sequestering with lncRNA *SNHG1* [166] and methylation-induced silencing [64].

Expression of miR-497-5p and mir-195-5p is down-regulated in the tumour tissue of patients with CRC compared to adjacent tissue or normal tissue, and high levels of these miRNAs have been associated with better survival (Appendix A).

Increased expression of miR-497-5p and/or miR-195-5p is associated with decreased cell proliferation, migration and EMT in the Lovo and SW480 cell lines in vitro and in vivo after their implantation into mice. This effect was prevented by sponging with lncRNA *SNHG1* [166]. 

High expression of miR-497-5p inhibits proliferation and invasion in CRC cell lines through targeting *IGF1R* [169], insulin receptor substrate 1 (*IRS1*), which influences IGF1R signalling [170], protein tyrosine phosphatase non-receptor type 3 (*PTPN3*), which is involved in the regulation of cell growth and differentiation [171], and kinase suppressor of ras 1 (*KSR1*), which induces the Raf/MED/ERK pathway and via its influence oncogenic transformation as well [172]. The target genes of miR-497-5p are also members of the Fos gene family, i.e., FOS-like 1, AP-1 transcription factor subunit (*FOSL1*), which is involved in the promotion of metastasis in CRC [173] (Appendix A).

Several studies indicate that miR-195-5p can increase the sensitivity of 5FU-resistant SW620 and HT-29 cell lines to chemotherapy by targeting transcriptional regulators, notch receptor 2 (*NOTCH2*) and recombination signal binding protein for immunoglobulin kappa J region (*RBPJ*) involved in the Notch signalling pathway, both of which are necessary for the maintenance of stemness and chemoresistance in CRC cells [174]. A newly-identified effector of chemoresistance, glycerophosphodiester phosphodiesterase domain-containing 5 (*GDPD5*) (traditionally linked to glycerol metabolism), has been shown to be suppressed by miR-195-5p [175]. miR-195-5p also inhibits the proliferation of CRC cell lines through targeting fibroblast growth factor 2 (*FGF2*) and subsequent decreases in *CCNB1*, cyclin D2 (*CCND2*) and cyclin-dependent kinase 2 (*CDK2*) levels [176], as well as reduced cell viability by targeting *BCL2* [177]. Expression of miR-195-5p inhibits cell proliferation and invasion by targeting the genes encoding *NOTCH2* [178] and the NF-κB activator scaffold protein caspase recruitment domain family member 10 (*CARMA3*) [179]. 

On the other hand, it has been demonstrated that WEE1 G2 checkpoint kinase (*WEE1*) and checkpoint kinase 1 (*CHEK1*) are targeted by miR-195-5p, which promotes the acquisition of drug resistance to 5FU in HCT-116 cells [180].

## 4. Regulation of Expression of Identified Clusters

A comprehensive analysis of miRNA clusters down-regulated in CRC revealed that they are predominantly located in a host gene sequence, in gene introns in most cases. None of the analysed clusters is situated on a sex chromosome. In spite of the generally accepted assumption that intron-derived miRNAs are transcribed from their host gene [11,18,20], it has recently been determined that more than 30% of intronic miRNAs possess upstream regulatory elements [21,22]. This finding is in complete agreement with our study because, with the exception of miR-15/16, TSSs independent of the host gene were found for all clusters (Table 1), which implicates cluster-specific transcriptional regulation. Moreover, we described regulation mediated by lncRNAs for at least one member of each cluster, which constitutes an additional level of miRNA regulation.

In spite of the complexity of miRNA control, it is of interest that all selected clusters show decreased expression, although, in some cases, there is still the need for experimental evidence to achieve a complete conclusion. Moreover, oncostatic functions are linked to the reviewed miRNAs, and high expression is usually associated with better patient survival, which is of interest since the abovementioned miRNAs are regulated differently. One uniform explanation for decrease in their expression in CRC tumours can be based on their active transport from cancer cells, as has been described previously [167]. However, this assumption needs to be experimentally validated. miRNA clusters that demonstrate tumour-suppressive functions have the potential to become a multi-target therapeutic tool to manipulate the amplification of several tumour-suppressive miRNAs by one promoter.

A major limitation of this study is an insufficient amount of information about the transcriptional regulation of the host gene, cluster and cluster members. In several cases, miRNA members of a particular cluster have been reported to be co-expressed; however, there is not always sufficient data to correlate the expression of clusters with their host genes in CRC tissue. Therefore, there is a lack of evidence supporting the notion that miRNA expression is regulated by host gene promoters. Moreover, post-transcriptional regulation and turnover, which can differ for particular miRNAs, probably influence the effective levels of miRNAs [11]. 

## 5. Target Genes and Functions of Identified Clusters

All miRNAs identified by literature search in this study execute their function via the broadly-conserved seed sequence and, with the exception of miR-194, belong to families containing more than one miRNA (Appendix A). There is experimental evidence supporting interference of miRNAs with decreased expression with more than 100 genes stimulating tumour progression in CRC (Appendix A). The most targeted genes were anti-apoptotic *BCL2* silenced by miRNAs from five clusters and four families and pro-angiogenic *VEGFA* regulated by four clusters from four families. The family containing clusters miR-497/195 and miR-15/16 targets 29 oncogenes, which is the highest amount for the families involved in this study. Considering the number of targeted genes, the most influential cluster is miR-143/145, targeting 26 genes, followed by miR-206/133b silencing 16 genes and miR-15/16 and miR-215/194-1 targeting 15 genes 

Experimental evidence validating in silico predictions of miRNA interactions with their target genes are most probably not complete, since miRNAs belonging to the same family rarely share the same target genes (Appendix A). In spite of incomplete experimental evidence, it is possible to implicate major directions in which clusters with decreased expression in CRC execute their oncostatic functions (Figure 1; Appendix A, GO analysis). GO analysis performed with use of the PANTHER Classification System showed that most of the target genes are classified as gene-specific transcriptional regulators, protein-modifying enzymes and cytoskeletal proteins. Classification according pathways showed that the most influenced pathways were angiogenesis, inflammation mediated by chemokine and cytokine and apoptosis signalling pathways (Appendix A, GO analysis).

The involvement of cluster miR-15/16 in CRC regulation was expected, in spite of the fact that the tumour-suppressive role of this cluster was originally discovered in chronic lymphocytic leukaemia [124]. As this cluster targets many genes, its effects are diverse, involving cell cycle control, apoptosis, cell migration and chemo- and radiosensitivity induction (Appendix A). However, it is surprising that cluster miR-143/145, known for its enrichment in vascular tissue and role in early heart morphology and vascular smooth muscle cell differentiation [61], shows such strong pleiotropic effects in CRC [138,139]. Cluster miR-206/133b, known mainly for its muscle-specific expression and capacity to regulate muscle development, function and regeneration, has been shown to be involved in regulation of CRC, mainly via its developmentally-active target genes (e.g., *NOTCH3* and *HOXA9*). Similarly, cluster miR-302b/302c/302a/302d/367 is involved in the control of pluripotency, self-renewal and reprogramming in human embryonic stem cells [162], which although rarely studied with respect to CRC, has been found to be especially useful in the induction of sensitivity to chemotherapy [24,163]. It seems that, although down-regulated clusters show tumour-suppressive functions via a wide range of target genes, it is possible to observe specific effects in some of them. As transcription of several miRNAs can be induced by one TSS, eventually, two TSSs could be used to activate two tumour-suppressive clusters with complementary functions to achieve better outcomes.

## 6. Conclusions

Taken together, down-regulated clusters are in most cases localised within genes (usually within introns) and fulfil tumour-suppressive roles. In spite of growing evidence about the regulation of miRNA transcription, a unifying mechanism of their decreased expression is not available. Even if a miRNA is localised inside a host gene and is transcribed along with it, there can still be several TSSs that can regulate miRNA transcription under specific conditions. Information about the transcriptional regulation of miRNA clusters has excellent potential to be used in translational research. This assumption is supported by the presence of several clusters that share important properties—their expression is decreased in CRC and they show oncostatic capacity. Better knowledge about the transcriptional regulation of tumour-suppressive clusters in CRC may, in the future, open the possibility of multi-target therapeutic manipulation executed via the activation of one promoter.

## Figures and Tables

**Figure 1 ijms-21-04633-f001:**
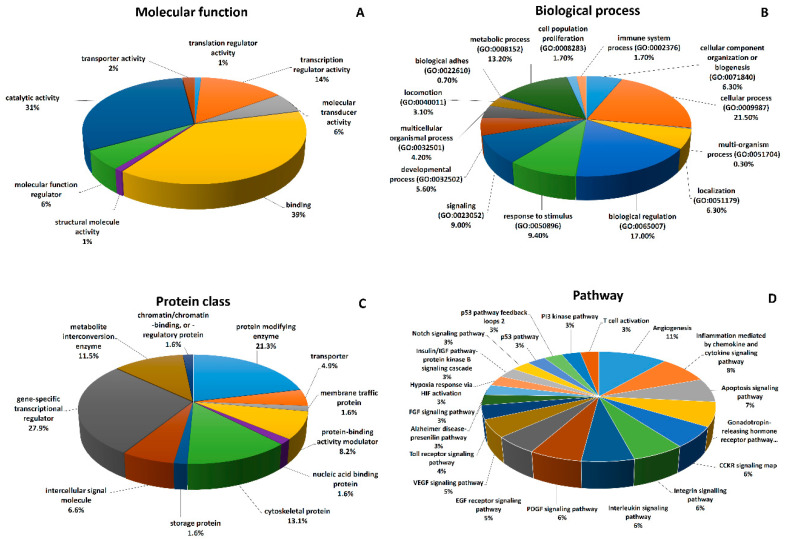
Gene ontology enrichment analysis of target genes of down-regulated miRNA clusters. (**A**) Classification according to the molecular function of genes, (**B**) classification according to biological processes in which target genes are involved, (**C**) classification according to protein class and (**D**) classification according pathways used in target gene signalling (first 20 most abundant pathways are plotted).

**Table 1 ijms-21-04633-t001:** Host gene, location and transcription start site of clusters down-regulated in colorectal cancer (CRC).

Cluster Chromosome	Host Gene [13] RNA Class	Cluster Position	Regulation of Cluster Transcription [45]
miR-100/let-7a-2/miR-125b-1 *Chr11*	*MIR100HG* ncRNA	intron	Predicted transcription start site (TSS) miR-100/let-7a-2, miR-125b-1 [46], whole cluster co-expression [47], co-expression of miR-125b with MIR100HG [48]
miR-99a/let-7c *Chr21*	*MIR99AHG* ncRNA	intron	At least one host gene-independent TSS regulating the whole cluster [10,21], host promoter regulation [49], expression of miR-99a and let-7c correlate with MIR99AHG [50,51,52]
miR-99b/let-7e/miR-125a *Chr19*	SPACA6 protein coding *SPACA6R-AS* long ncRNA (lncRNA) antp.	mixed * exon	At least one host gene-independent TSS regulating the whole cluster [21,46,47,49], expression of miR-99b, let-7e and miR-125a correlate with SPACA6 [50,51,52]
miR-1-2/133a-1 *Chr18*	MIR133A1HG lncRN A MIB1 protein coding antp.	exon intron	Host gene-independent TSS for the whole cluster [47], expression of miR-1-2 and miR-133a-1 does not correlate with MIB1 [50,51,52]
miR-1-1/133a-2 *Chr20*	*MIR1-1HG* unknown	mixed ^#^	Host gene-independent TSS for the whole cluster [53], expression of miR-1-1 and miR-133a-2 do not correlate with MIR1-1HG [50,51,52]
miR-206/133b *Chr6*	miR-206 miR-133b *LINCMD1* ncRNA	intergenic intron	Host gene-independent TSS for the whole cluster [47,49,54]
miR-192/194-2 *Chr11*	*MIR194-2HG* lncRNA	mixed *	At least one host gene-independent TSS regulating the whole cluster [49,55], promoter regulating miR-194 [56]
miR-215/194-1 *Chr1*	IARS2 protein coding	intron	At least one independent TSS regulating the whole cluster [53,57,58], TSS for miR-215 [59], expression of miR-194-1 correlates with IARS2, expression of miR-215 does not correlate with IARS2 [50,51,52]
miR-15a/16-1 *Chr13*	DLEU2 lncRNA	mixed *	DLEU2 promoter [49], expression of miR-15a correlates with DLEU2, expression of miR-16-1 does not correlate with DLEU2 [50,51,52]
miR-15b/16-2 *Chr3*	SMC4 protein coding *TRIM59-IFT80* lncRNA antp.	intron intron	SMC4 promoter [21,49], expression of miR-16-2 and miR-15b do not correlate with expression of SMC4 [50,51,52]
miR-143/145 *Chr5*	*CARMN* lncRNA	mixed ^#^	Identification of independent TSS for the whole cluster [47], correlation with host gene expression [60], knock-down of *CARMN* decreases expression of miR-143 and -145 [61], promoter regulation of miR-145 expression [62]
miR-302b/302c/302a/ 302d/367 *Chr4*	*MIR302CHG* lncRN A LARP7 protein coding antp.	mixed ^#^ intron	At least one independent TSS regulating the whole cluster [53,63]
miR-497/195 *Chr17*	*MIR497HG* lncRNA	intron	At least one independent TSS regulating the whole cluster [49,64]

Host genes in parallel as well as antiparallel (antp.) DNA strands are shown. Chr, chromosome; mixed, located partially in intron, exon and/or intergenic region; *, intron or exon location depending on splice variant; #, intron/exon junction; TSS, transcription start site mediating regulation independent from host gene; lncRNA, long non-coding RNA; ncRNA, non-coding RNA; HG, host gene, CARMN, cardiac mesoderm enhancer-associated non-coding RNA; DLEU2, deleted in lymphocytic leukaemia 2; IARS2, isoleucyl-tRNA synthetase 2, mitochondrial; LINCMD1, long intergenic non-protein coding RNA, muscle differentiation 1; MIB1, mindbomb E3 ubiquitin protein ligase 1; SMC4, structural maintenance of the chromosomes protein 4; SPACA6, sperm acrosome associated 6; TRIM59-IFT80, tripartite motif-containing 59 and intraflagellar transport 80.

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
