# Peer review of "miRNA Clusters with Down-Regulated Expression in Human Colorectal Cancer and Their Regulation"

_ijms, 2020, doi:10.3390/ijms21134633_

Round 1

Reviewer 1 Report

Review of the manuscript ijms-812545.

The paper by Pidikova et al is a detailed overview on clusters of miRNAs that have a role in CRC. Authors presented a large quantity of the reported evidence showing a general down regulation of these miRNAs in CRC and an important regulation mediated by specific lncRNAs and circRNAs.

I would like to congratulate with the authors for the large work done: it is not easy to put together data from such heterogeneous studies and make a collective summary.

Despite this, I would like to emphasize that the work needs a revision since it is not easy to read and follow. This is due partly to the large amount of data presented with a tendency of mixing some parts and to create a sort of “shopping list” for each miRNA. Along the text there are some parts with the description of the specific miRNA and other parts with just the not mature miRNAs (without -3p or -5p). This is quite confusing. On the other hand, there are several parts that show heavy sentences not all the time correct (see for example: lines 325-330 or lines 422-425). Moreover, some introductory part to the clusters are redundant (see line 246-247.. the sentence is not giving any info..)

Main points:

  • I would to invite the authors in keeping more consistency in the text: the use of acronyms is a bit random (see line 187 and 197 for MCL1 or lines 93-94 for lncRNAs). Please use acronyms when necessary and list not only the full name of genes but also their symbols. Moreover, in the text genes and RNAs should be in Italics (to help to distinguish better proteins and genes) and I would invite authors in being consistent in the text because some genes are introduced with their name in extenso, while other time this is not done.
  • I would recommend also a revision of the manuscript by a native English speaker because there some errors (line 42 instead of “recognized as” it should be used “recognized by”; line 34 “This category is composed from miRNAs..” Change from with by etc).
  • Authors should describe better the aims of the study: why they wanted to search for clusters? Why in CRC?? These points are not emerging after reading the whole manuscript especially the focus on clusters as produced by the authors.
  • The definition of passenger strand and * is obsolete (lines 68-69). And the sentence is in contrast with the following one (lines 69-70).
  • The question of definition of clusters as done in the manuscript is another issue the authors should explain or clarify better. I don` t like so much as they put together more than one clusters. Authors did not give a personal definition of the clusters presented (why for example to put together miR-100/let-7a-2/miR-125b-1, miR-99a/let-7c and miR-99b/let-7e/miR-125a? Because they have some families of miRNAs in common? ) I don` t see the rationale and even looking for help on miRbase it does not help because miRbase does not use such a definition of cluster (for miRbase these are 3 separated clusters! The problems emerged in situation like at line 206. Please consider the sentence “In CRC transcription, mir-1-1 and miR-133a have been shown to be regulated by DNA methylation” Besides the use of the wrong terms, it is confusing the use of one of the copy of miRNA in one cluster (miR-1-1 from chromosome 20) and the name of the mature miRNA generated by 2 clusters (miR-133a  is the mature version and without the chromosome of origin derived from miR-133a-1 and miR-133a-2). The correct form should be “miR-1 and miR-133a”
  • The main message that reader can take is a general downregulation of the clusters presented (see abstract and lines 466 and later on), which is widely discussed in the conclusive part of the paper. However, I would like to remind that authors actively selected only clusters downregulated in CRC as stated at lines 128-130 and line 134. So I am a bit confused on the discrepancy between the initial part of the work and the conclusive discussion.
  • One of the limitation given by the authors is “..an insufficient amount of information about transcriptional regulation of the host gene, cluster and..”. I don` t agree with the whole part since the authors could easily access to publicly available database such as TCGA and verify themselves the concordance of expression of miRNAs in intronic clusters and their host genes. Why the authors did not test this?
  • The part relative to the targets of miRNA cluster was done by the authors (lines 436-449)? Is it from the literature? It is not clear.

Minor:

  • I would chose a better list of keywords instead the full list of miRNAs.
  • The first part of Introduction (lines 29-36) is a deviation from the rest of the manuscript. The focus is microRNAs and the whole part on ncRNAs is just out of the topic.
  • Reference 17 is too old for the description of miRNAs in CRC.
  • The sentence at lines 169-170 is exactly the same at lines 132-133. It could be that it is the legend of the table but it is not clear.
  • I found not correct the extensive use in the text of terms like “In CRC transcription” (line 206) or “CRC expression of miR..” (line 326). Also “regulation by DNA methylation” (line 206-207) is too general and ambiguous.
  • Jumping from studies on cell lines to xenographs or tissue expression levels creates confusion..
  • Consider the sentence at lines 446-449. It is contradictory. Can you reformulate it?

Author Response

Dear reviewer,

thank you very much for your valuable time and all comments and suggestions. We accepted all of them and here we provide detailed list of changes. We hope, you will find revised MS acceptable.

The paper by Pidikova et al is a detailed overview on clusters of miRNAs that have a role in CRC. Authors presented a large quantity of the reported evidence showing a general down regulation of these miRNAs in CRC and an important regulation mediated by specific lncRNAs and circRNAs.

I would like to congratulate with the authors for the large work done: it is not easy to put together data from such heterogeneous studies and make a collective summary.

Despite this, I would like to emphasize that the work needs a revision since it is not easy to read and follow. This is due partly to the large amount of data presented with a tendency of mixing some parts and to create a sort of “shopping list” for each miRNA.

- we apologise for complicated sentence structure, we did our best to improve readability of MS and let control grammar and sentence structure again in professional editing company (please, see attached certificate)

Along the text there are some parts with the description of the specific miRNA and other parts with just the not mature miRNAs (without -3p or -5p). This is quite confusing.

- thank you for the comment. We consider to be an important advantage of recent MS that we refer to particular miRNA strand as many articles do not provide this information explicitly (you must check primer and probe sequences etc. to learn what was done). Recently it has been accepted that in some cases passenger strand can also exerts some regulatory function [4,8,9], therefore, referring to  particular strand becomes even more relevant.

In recent version of MS we supplemented information about type of strand in all experimental studies where expression was measured. Only in case that the whole cluster is mentioned, or it was really obvious that both strands are influenced by treatment, e.g. methylation, we, naturally, do not refer to particular strand.

On the other hand, there are several parts that show heavy sentences not all the time correct (see for example: lines 325-330 or lines 422-425).

- thank you for the comment, heavy sentences were re-formulated, MS was proof edited by professional agency again

lines 325-330 in former version

Bicistrone miR-143/-145 are negatively regulated by Ras responsive element binding protein 1 [119], and the expression of both miRNAs is also suppressed via EGFR [120]. CRC expression of miR-145-5p is also controlled by methylation [121] and snail family transcriptional repressor 1 [122] and sponged by circRNA CIRC_001569 [123], snoRNA SNHG1 [124], lncRNA SOX21-AS1 [125], lncRNA CASC15 [126], circRNA PVT1 [127] and lincRNA-ROR [128]. miR-143-3p is sequestered by lincRNA UCC [129], ceRNA PART-1 [130] and lncRNA OECC in CRC [131].

new text 343-349

Bicistronic miR-143/145 is negatively regulated by Ras responsive element binding protein 1 (RREB1) [138], and the expression of both miRNAs is suppressed via EGFR [139]. The core promoter region of miR-145 is regulated by histone methylation in CRC cell lines [62] and snail family transcriptional repressor 1 (SNAI1) [140]. miR-145-5p is sponged by circRNA CIRC_001569 [141], snoRNA SNHG1 [142], lncRNA SOX21-AS1 [143], lncRNA CASC15 [144], circRNA PVT1 [145] and lincRNA-ROR [146]. miR-143-3p is sequestered by lincRNA UCC [147] and ceRNA PART-1 in SW620 cells [148].

lines 422-425 in former version

Importantly, as for all clusters, transcriptional regulation that influences all members of the clusters has been described, and at the same time, they demonstrate tumour-suppressive functions. As a result, there is a promise of multi-target therapeutic manipulation by amplification of several tumour-suppressive miRNAs by one promoter.

new text 447-540

miRNAs clusters that demonstrate tumour-suppressive functions have the potential to become a multi-target therapeutic tool to manipulate the amplification of several tumour-suppressive miRNAs by one promoter.

Moreover, some introductory part to the clusters are redundant (see line 246-247.. the sentence is not giving any info..)

- thank you for the comment, sentence was re-formulated

lines 246-247 in former version

The miR-192/miR-194-2 and miR-215/miR-194-1 paralogues are located on different chromosomes (Table 1). Mature sequences of miR-192-5p and miR-215-5p differed only by 2 nt [82].

new text 447-540

Clusters miR-192/194-2 and miR-215/194-1 are located on different chromosomes (Table 1) and consist of miR-194 and miR-192 or miR-215, which differ by only 2 nt [56].

Main points:

I would to invite the authors in keeping more consistency in the text: the use of acronyms is a bit random (see line 187 and 197 for MCL1 or lines 93-94 for lncRNAs). Please use acronyms when necessary and list not only the full name of genes but also their symbols.

- thank you for the comment, acronyms were controlled, symbols of genes according HUGO Gene Nomenclature were provided

Moreover, in the text genes and RNAs should be in Italics (to help to distinguish better proteins and genes) and I would invite authors in being consistent in the text because some genes are introduced with their name in extenso, while other time this is not done.

- thank you very much for the comment. Manuscript was a little bit influenced by way how gene and protein names were given in original papers. Moreover, in many cases the authors studied effect of miRNA not only on mRNA but also protein expression (or both). Genes and RNAs were written in italic capital letters and name of protein in regular capital letters BUT referring to RNA or protein was dependent on the original context. Therefore, in some cases it looked like we do not make difference between RNA and protein and text was confusing.

- in recent version we use only HUGO Gene Nomenclature (without regard on alias used in the original paper). When we describe transcriptional factors we use regular capital letters and when describe target genes we use italic capital letters. We first provide whole name of target gene and then symbol.

We followed instruction on published on page:

https://www.biosciencewriters.com/Guidelines-for-Formatting-Gene-and-Protein-Names.aspx#:~:text=General%20formatting%20and%20writing%20guidelines&text=In%20general%2C%20symbols%20for%20genes,italicized%20(e.g.%2C%20IGF1).

Authors should describe better the aims of the study: why they wanted to search for clusters? Why in CRC?? These points are not emerging after reading the whole manuscript especially the focus on clusters as produced by the authors.

- thank you very much, for the idea, reasoning was added into chapter 1. introduction lines 31-34

Colorectal cancer (CRC) is the fourth most common cancer worldwide with high mortality [1]. In spite of progress in CRC diagnostics and the determination of patient prognosis, there is still a need for improvement. During last two decades, miRNAs have been frequently discussed as a potential tool for the assessment of cancer progression [2].

and chapter 2. Regulation of miRNA expression lines 95-102

In spite of huge progress in miRNA measurement [2] there remain inconclusive and contradictory results about miRNA up- or down-regulation in CRC, e.g. miR-204, miR-203, miR-200, miR-150, miR-142 etc. [17]. The reason for this may be due to differences in biological context or methods of miRNA evaluation; however, this inconsistency precludes these miRNAs from use as effective bio-tools. A recent review suggested that miRNAs organised in clusters may be more reliable biomarkers as they can share the same way of transcriptional regulation. We focused on down-regulated clusters, as their levels are less likely to be masked by cell fragmentation due to cell death.

The definition of passenger strand and * is obsolete (lines 68-69). And the sentence is in contrast with the following one (lines 69-70).

- thank you for the idea, text was replaced by:

 “The second miRNAs strand is present in the cytoplasm at much lower concentration compared with the guide strand; it is denoted with an asterisk and, in spite of its low concentration, can be to some extent also incorporated into the RISC [4,8,9].“ lines 67-69

The question of definition of clusters as done in the manuscript is another issue the authors should explain or clarify better. I don` t like so much as they put together more than one clusters. Authors did not give a personal definition of the clusters presented (why for example to put together miR-100/let-7a-2/miR-125b-1, miR-99a/let-7c and miR-99b/let-7e/miR-125a? Because they have some families of miRNAs in common? )

- thank you for the comment. Actually, an idea of introducing clusters in this way was not ours. We followed an example of cluster description in a paper (Emmrich et al., Genes Dev. 2014, 79 citations in Scopus on June 5, 2020) since we believed, that it would increase readability of MS. However, after another literature search we agree that this is not a usual way of cluster listening, it is difficult to clearly define it and therefore we abandoned idea of cluster “clustering” in the whole MS. Instead of it we provided in the abstract, chapter titles and all tables full names of all clusters or their members (depending on context). Since clusters mentioned in particular subchapters are really functionally related as they contain members from the same families and their role in a carcinogenesis (and experimental evidence supporting it) is also similar we maintained structure of chapter 3 with 7 subchapters. If we describe functionally related clusters in separate subchapters it would lead to referring of the same studies repetitively and it would significantly decrease readability of MS for the reader.

I don` t see the rationale and even looking for help on miRbase it does not help because miRbase does not use such a definition of cluster (for miRbase these are 3 separated clusters! The problems emerged in situation like at line 206. Please consider the sentence “In CRC transcription, mir-1-1 and miR-133a have been shown to be regulated by DNA methylation” Besides the use of the wrong terms, it is confusing the use of one of the copy of miRNA in one cluster (miR-1-1 from chromosome 20) and the name of the mature miRNA generated by 2 clusters (miR-133a  is the mature version and without the chromosome of origin derived from miR-133a-1 and miR-133a-2). The correct form should be “miR-1 and miR-133a”

- than you for correction, text was corrected accordingly, line 217

The main message that reader can take is a general down regulation of the clusters presented (see abstract and lines 466 and later on), which is widely discussed in the conclusive part of the paper. However, I would like to remind that authors actively selected only clusters downregulated in CRC as stated at lines 128-130 and line 134. So I am a bit confused on the discrepancy between the initial part of the work and the conclusive discussion.

- thank you for the comment. The main message of the MS is that information about transcriptional regulation of miRNAs clusters has much higher potential to be used in translational research than it happens recently. This assumption is supported by listing of several clusters that share the same basic properties: they are down-regulated and show oncostatic capacity. Down-regulated clusters with this features can become promising multi-targets for therapeutic manipulation. This idea was emphasize in chapter 4. Conclusions.

One of the limitation given by the authors is “..an insufficient amount of information about transcriptional regulation of the host gene, cluster and..”. I don` t agree with the whole part since the authors could easily access to publicly available database such as TCGA and verify themselves the concordance of expression of miRNAs in intronic clusters and their host genes. Why the authors did not test this?

- thank you very much for the comment. As it is obvious from table below even such a progressive tool as a TCGA repository does not contain all data for correlation analysis between miRNAs and host genes. Moreover, most of studies found in original literature search are not listed in TCGA repository et all. TCGA repository contains mainly information from NGS screening studies and this technique is not primarily designed to perform correlation between mRNA and miRNA, e.g. the authors usually do not pay attention to particular splicing variant of host genes as general screening is the aim. However, in some cases correlations could be calculated. This information was added into Table 1.

 - this table shows comparison between original search and TCGA search

Cluster

Host gene [13]

Cluster position

Regulation of cluster transcription [81]

TCGA database

Chromosome

RNA class

miR-100/let-7a-2/miR-125b-1

MIR100HG

intron

Predicted TSS miR-100/let-7a-2, miR-125b-1 [167], whole cluster co-expression [168], co-expression of miR-125b with MIR100HG [169]

no data

Chr11

ncRNA

miR-99a/let-7c

MIR99AHG

intron

At least one host gene independent TSS regulating the whole cluster [18,10], host promoter regulation [170]

miR-99a and let-7c correlate with host gene

Chr21

 ncRNA

miR-99b/let-7e/miR-125a

SPACA6

mixed*

At least one host independent TSS regulating the whole cluster [18,167,168,170]

miR-99b, let-7e and miR-125a  correlate with SPACA6R-AS

Chr19

protein coding

SPACA6R-AS

exon

lncRNA antp.

miR-1-2/133a-1

MIR133A1HG

exon

Host gene independent TSS for the whole cluster [168]

no data for MIR133A1HG

Chr18

lncRNA

miR-1-2 and

miR-133a-1 do not correlate with MIB1

MIB1

intron

protein coding antp.

miR-1-1/133a-2 Chr20

MIR1-1HG

mixed#

Host gene independent TSS for the whole cluster [171]

miR-1-1 and miR-133a-2 do not correlate with MIR1-1HG

unknown

miR-206/133b

miR-206

intergenic

Host gene independent TSS for the whole cluster [168,170,172]

no data

Chr6

miR-133b

intron

LINCMD1

ncRNA

miR-192/194-2

MIR194-2HG

mixed*

At least one host gene independent TSS regulating the whole cluster [170,173], promoter regulating miR-194 [82]

no data for MIR194-2HG

Chr11

lncRNA

miR-215/194-1

IARS2

intron

At least one independent TSS regulating the whole cluster [83,171,174]

miR-194-1 correlates with IARS2, miR-215 does not correlate with IARS2

Chr1

protein coding

miR-15a/16-1

DLEU2

mixed*

DLEU2 promoter [170]

miR-15a correlates with DLEU2, miR-16-1 does not

Chr13

lncRNA

miR-15b/16-2

SMC4

intron

SMCA4 promoter [18,170]

miR-15b and miR-16-2 do not correlate with SMC4

Chr3

protein coding TRIM59-IFT80

lncRNA antp.

intron

miR-143/145

CARMN

mixed#

Identification of independent TSS for the whole cluster [168], correlation with host gene expression [175], knock-down of CARMN decreases expression of miR-143 and -145 [166], promoter regulation of miR-145 expression [121]

no data

Chr5

lncRNA

miR-302b/302c/302a/302d/367

MIR302CHG

mixed#

At least one independent TSS regulating the whole cluster [142,171]

no data

Chr4

lncRNA

 LARP7

intron

protein coding antp.

miR-497/195

MIR497HG

intron

At least one independent TSS regulating the whole cluster [153,170]

no data

Chr17

lncRNA

A sentence: “a correlation of the expression of clusters with their host genes has rarely been studied in CRC was replaced by“ a sentence “there is not always sufficient data to correlate the expression of clusters with their host genes in CRC tissue” line 453-454

The part relative to the targets of miRNA cluster was done by the authors (lines 436-449)? Is it from the literature? It is not clear.

- thank you for the comment, this two paragraphs are result of our analysis of available literature. Text was reformulated and we also added references to particular supplementary data to make it clearer.

original text:

All miRNAs identified in this study execute …

Experimental evidence validating in silico predictions of miRNA interactions with their target genes are most probably not complete, since miRNAs belonging to the same family rarely share the same target genes (Table 3).This finding is, of course, partially the result of specific miRNA regulation in CRC cells; however, the expected overlay between family members and targeted genes is higher than recently observed. In spite of this, it is possible to implicate major directions in which down-regulated clusters execute their oncostatic functions.

recent text:

All miRNAs identified by literature search in this study execute their … line 458

Experimental evidence validating in silico predictions of miRNA interactions with their target genes are most probably not complete, since miRNAs belonging to the same family rarely share the same target genes (Tables S1 and S3). In spite of incomplete experimental evidence, it is possible to implicate major directions in which down-regulated clusters execute their oncostatic functions (Figure 1, S4 - GO analysis). lines 468-472

Minor:

I would chose a better list of keywords instead the full list of miRNAs.

- keywords were changed and miRNAs were omitted from list of keywords

A recent list of key words is: proliferation, apoptosis, chemoresistance, survival, long ncRNA, methylation, angiogenesis, cell adhesion

The first part of Introduction (lines 29-36) is a deviation from the rest of the manuscript. The focus is microRNAs and the whole part on ncRNAs is just out of the topic.

- I believe I understand you point, however, we have to introduce miRNA as a group of regulatory molecules somehow. Lines 30-36 were deleted not to distract attention from main class of ncRNAs

original text:

MicroRNAs (miRNAs) belong to a large family of non-coding RNAs (ncRNAs). More than 90% of the human genome is devoted to the synthesis of ncRNAs. This is a surprisingly high number considering that ncRNAs do not provide templates for protein production [1]. Instead, ncRNAs fulfil other functions according to their specific sub-classes. There are several ways to classify ncRNAs. The most common way of classifying ncRNAs is according their lengths. ncRNAs with length less than 200 nucleotides (nt) are named small ncRNAs. This category is composed from miRNAs, small interfering RNAs (siRNAs), piwi-interacting RNAs (piRNAs), small nucleolar RNAs (snoRNAs), small nuclear RNAs (snRNAs) and transfer RNA-derived small RNAs (tsRNAs) [1,2].

recent text:

MicroRNAs (miRNAs) belong to a large family of non-coding RNAs (ncRNAs). The average length of miRNAs is only 22 nt. The canonical pathway … lines 35-36

Reference 17 is too old for the description of miRNAs in CRC.

- reference 17 is linked to paper Baskerville S, Bartel DP. Microarray profiling of microRNAs reveals frequent co-expression with neighboring miRNAs and host genes. RNA. 2005 11(3):241-7 that is cited by 1068 references and it is impossible to ignore it. Even in 2019 this paper was referred by 42 articles without auto-citations that indicates huge impact and persisting influence on field development.

We omitted ref. no. 17 in Table 1 as we have, thanks to you, more recent measurement of correlation now.

The sentence at lines 169-170 is exactly the same at lines 132-133. It could be that it is the legend of the table but it is not clear.

- the sentence is really part of Table 1 legend. We would like to preserve this information in the legend as way how cluster is defined can vary between studies and in this way definition is easily accessible for reader

I found not correct the extensive use in the text of terms like “In CRC transcription” (line 206) or “CRC expression of miR..” (line 326). Also “regulation by DNA methylation” (line 206-207) is too general and ambiguous.

- thank you for the comment, text was reformulated

original text:

In CRC transcription, mir-1-1 and miR-133a have been … was changed into: miR-1 and miR-133a expression line 217

CRC expression of miR of miR-145-5p … this paragraph was completely changed lines 343-349

Regulation by DNA methylation … was changed into… miR-1 and miR-133a expression is silenced by DNA hypermethylation line 217

We corrected also similar formulations in text.

Jumping from studies on cell lines to xenographs or tissue expression levels creates confusion.

- mentioning of xenografts was omitted from the text

Consider the sentence at lines 446-449. It is contradictory. Can you reformulate it?

- part of sentence was omitted from text, a critical sentence was deleted.

original text:

Experimental evidence validating in silico predictions of miRNA interactions with their target genes are most probably not complete, since miRNAs belonging to the same family rarely share the same target genes (Table 3).This finding is, of course, partially the result of specific miRNA regulation in CRC cells; however, the expected overlay between family members and targeted genes is higher than recently observed. In spite of this, it is possible to implicate major directions in which down-regulated clusters execute their oncostatic functions.

recent text:

Experimental evidence validating in silico predictions of miRNA interactions with their target genes are most probably not complete, since miRNAs belonging to the same family rarely share the same target genes (Tables S1 and S3). In spite of incomplete experimental evidence, it is possible to implicate major directions in which down-regulated clusters execute their oncostatic functions (Figure 1, S4 - GO analysis). lines 468-472

Reviewer 2 Report

The manuscript „miRNA Clusters with Down-Regulated Expression in Human Colorectal Cancer and Their Regulation“ summarizes the current knowledge about the miRNAs’ clusters with decreased expression in colorectal cancer (CRC), their transcriptional and post-transcriptional control, association with patients’ survival as well as their tumor suppressive properties through targeting multiple oncogenic pathways. The whole manuscript is very comprehensive, the text is well written and there are only few mistakes to be corrected. However, I would suggest to add some schematic pictures or graphs to increase the readability of the manuscript.

Minor corrections:

1) There is a disunity in usage of miRNA/miRNAs singular/plural during the manuscript.

2) Table 1 – the headings for miR-143/145, miR-302, and miR-497 clusters are missing.

3) Disunity in usage of abbreviations vs. whole names of target genes .

4) In Conclusion, the authors state that miR-497/-195 and miR-15/-16 clusters target 33 tumour-suppressive genes, however, these miRNAs target predominantly important oncogenes (similarly for other clusters). In addition, these genes are experimentally confirmed, nevertheless, the number is probably much more higher.

5) Conclusion, line 454 – probably early heart morphology (instead of heard)

6) S1 Table – there is no explanation for italics in case of miR-194-5p

Further recommendations:

1) I would suggest to remove Table 2 (or put it as supplementary) as this Table says nothing new – we know that all described miRNAs are down-regulated in CRC, thus they are supposed to function as tumour suppressors and their high levels should be associated with better survival. I know there are some exceptions (especially in case of miR-15/16 family), but I suggest to include this information into the text. In addition, some of the references (50, 88, 134) are used in both columns concerning the tumour suppressive or oncogenic role of miRNAs - that is a little bit strange and confusing.

2) Instead of Table 3, I would recommend to use some graphic presentation of the data – e.g. particular signalling pathways with target genes regulated by described miRNA clusters. Based on the Table, it is very difficult to find any connection between individual miRNA and its role in cancerogenesis.  

Author Response

Dear reviewer,

thank you very much for positive assessment and constructive comments. We did our best to correct text to fulfil required changes and agree that suggested modifications improved quality of MS. We hope that MS in recent form is acceptable for publication in IJMS.

1) There is a disunity in usage of miRNA/miRNAs singular/plural during the manuscript.

- thank you for the comment, we switched to plural everywhere where it was possible

2) Table 1 – the headings for miR-143/145, miR-302, and miR-497 clusters are missing.

- table was completely reorganized to fulfil comments of referee 1 so all headings were deleted. In this way inconsistency was removed, we hope, that this solution is acceptable.

3) Disunity in usage of abbreviations vs. whole names of target genes.

- thank you for the comment, all target genes names and symbols were controlled to fulfil rules of HUGO Gene Nomenclature. We always introduce name of gene in full and then provide abbreviation.

4) In Conclusion, the authors state that miR-497/-195 and miR-15/-16 clusters target 33 tumour-suppressive genes, however, these miRNAs target predominantly important oncogenes (similarly for other clusters). In addition, these genes are experimentally confirmed, nevertheless, the number is probably much more higher.

- thank you for correcting this phrase, it was a overtyping, of course target genes mentioned in our MS are mostly oncogenes, a statement was changed to “The family containing clusters miR-497/-195 and miR-15/-16 target 29 oncogenes”

5) Conclusion, line 454 – probably early heart morphology (instead of heard)

- many thanks you for the comment, text was corrected

6) S1 Table – there is no explanation for italics in case of miR-194-5p

- thank you for the comment, italics was changed to regular font

Further recommendations:

1) I would suggest to remove Table 2 (or put it as supplementary) as this Table says nothing new – we know that all described miRNAs are down-regulated in CRC, thus they are supposed to function as tumour suppressors and their high levels should be associated with better survival. I know there are some exceptions (especially in case of miR-15/16 family), but I suggest to include this information into the text.

- table 2 was put as supplementary, in this way information about context dependent oncogenic potential of miR-15/16 family was preserved. Moreover, Table 2 seems to be obvious only when it is clearly presented, but it was not so obvious (at least to us) before we performed a complex search of the literature. We hope it is acceptable to keep it in supplementary files.

In addition, some of the references (50, 88, 134) are used in both columns concerning the tumour suppressive or oncogenic role of miRNAs - that is a little bit strange and confusing.

- ref 50 (in new version of MS 73),  88 (107) and 134 (151) refer to papers showing that in some cases miRNAs show divergent features. This is not so unexpected as miRNAs always target more than one gene. However, to make interpretation of recent experimental studies clear we carefully controlled content of papers and removed ref. 107 and 151 from list of oncogenes and ref. 73 from list of tumour suppressors.

50/73 Zhang, X.; Ma, X.; An, H.; Xu, C.; Cao, W.; Yuan, W.; Ma, J. Upregulation of microRNA-125b by G-CSF promotes metastasis in colorectal cancer. Oncotarget 2017, 8, 50642-50654.

88/107 Cai, Y.; Yan, P.; Zhang, G.; Yang, W.; Wang, H.; Cheng, X. Long non-coding RNA TP73-AS1 sponges miR-194 to promote colorectal cancer cell proliferation, migration and invasion via up-regulating TGFα. Cancer Biomark 2018, 23:145-156.

134/151 Sathyanarayanan, A.; Chandrasekaran, KS.; Karunagaran, D. microRNA-145 downregulates SIP1-expression but differentially regulates proliferation, migration, invasion and Wnt signaling in SW480 and SW620 cells. J. Cell. Biochem. 2018, 119, 2022-2035.

2) Instead of Table 3, I would recommend to use some graphic presentation of the data – e.g. particular signalling pathways with target genes regulated by described miRNA clusters. Based on the Table, it is very difficult to find any connection between individual miRNA and its role in cancerogenesis. 

- table 3 was shifted to supplementary material and figure showing molecular functions, protein class, biological processes and pathway was created (Figure 1) and incorporated into MS. In addition, all result of GO analysis are provided in supplementary file to link particular target gene to particular process or pathway according GO. Tables S1 and S3 provide clue to link gene to cluster or family.

Round 2

Reviewer 1 Report

Review of the paper by Pidikova et al (IJMS)

The paper by Pidikova et al is a detailed overview on clusters of miRNAs that are usually downregulated in CRC and have a role in this type of cancer. Authors presented a large quantity of info and reported evidence showing a general downregulation of these miRNAs in CRC and an important regulation mediated by specific lncRNAs and circRNAs.

I have found a great implementation from the last version of the paper; however, I still find the manuscript very heavy to read and with high levels of similarity with this paper (DOI: 10.1002/wrna.1563) especially in the selection of clusters to be discussed.

I am aware that preparing such a big work it is not easy in a short and direct way. However, authors should consider the utility of the work for the scientific community and the possibility for readers to extrapolate significant messages after reading.

Main comments:

  • Lines 67-69: please modify the sentence in the most updated and correct way. The “second miRNA strand” is called passenger strand and the use of asterisk is obsolete (I think it has been changed already 2-3 miRBase versions ago). Please change with the designation of -3p and -5p arms.
  • Revise the sentence at lines 49-50. The function of Dicer is misleading here and not clear.
  • I found that manuscript is not well balanced in the distribution of the topics. There are almost 2 pages dedicated to function and maturation of miRNAs (which is a bit out of topic) and then regarding lncRNAs (more connected with the manuscript), authors dedicated a mini and very reductive paragraph (lines 87-93). Moreover, the conclusive part (4. Conclusions) it is almost 2 pages long, which I found excessive for a “conclusion”.
  • Regarding Conclusions section again, I suggest to authors to be more precise on the enrichment analyses performed.. The graphs relative to figure 1 from what type of enrichment analyses was done?? What is the program used? What is the list of genes used? The data are not even properly discussed and connected with the data reported for clusters.
  • I suggest a complete revision also of the initial part of chapter 2 (lines 95-102). The whole part does not make much sense to me.. “MiRNA measurement” is an improper term as well as “miRNA evaluation”. What the authors meant??Maybe they meant measurement of levels of expression? And miRNA extraction and methodology of measurement? Please, clarify it! And what does this sentence (“however, this inconsistency precludes these miRNAs from use as effective bio-tools”) mean ?
  • Please revise the text because the authors declared on lines 100-102 that they described only clusters downregulated in CRC. This concept has been repeated for all the clusters subsections and in the last part of the manuscript (I have counted at least 15 times!!). This will help to make the manuscript lighter to read.
  • I think authors should try to connect better the evidence reported for each clusters with the fact that those clusters (or better, the miRNAs in the clusters) were reported frequently downregulated in CRC. This will be the additional information for readers that will make the text something more than just a shopping list.

Minor:

-Line 84: change “into mature miRNAs” to “into a final mature miRNAs”

-lines 178-188: I think this part is relative to the Table but it looks like part of the text (and redundant to page 4)

Author Response

Dear reviewer,

thank you very much for positive assessment and constructive comments. We did our best to correct text to fulfil required changes and agree that suggested modifications improved quality of MS. We hope that MS in recent form is acceptable for publication in IJMS.

Comments and Suggestions for Authors

The paper by Pidikova et al is a detailed overview on clusters of miRNAs that are usually downregulated in CRC and have a role in this type of cancer. Authors presented a large quantity of info and reported evidence showing a general downregulation of these miRNAs in CRC and an important regulation mediated by specific lncRNAs and circRNAs.

I have found a great implementation from the last version of the paper; however, I still find the manuscript very heavy to read and with high levels of similarity with this paper (DOI: 10.1002/wrna.1563) especially in the selection of clusters to be discussed.

- we carefully read paper DOI: 10.1002/wrna.1563. This review entitled “Cluster miRNAs and cancer: Diagnostic, prognostic and therapeutic opportunities” by Kabekkodu and colleagues deals with 38 clusters with disturbed expression (up or down regulation) in 50 types of cancers. There is no special focus on colorectal cancer. There is an overlap only in 7 clusters out of 13 referred in recent review and out of these only 2 clusters (miR-1-1/133a-2 and miR-143/145, shown in bolt) are mentioned in relationship to colorectal cancer. We do not see any similarity between two articles that could question originality of our MS.

Clusters involved in recent MS that were not mentioned in DOI: 10.1002/wrna.1563 

miR-100/let-7a-2/miR-125b-1

miR-1-2/133a-1

miR-206/133b

miR-192/194-2

miR-15b/16-2

miR-497/195

List of miRNA clusters in
DOI: 10.1002/wrna.1563 

List of cancer types mentioned in

DOI: 10.1002/wrna.1563

1

Let-7c/miR-99a/miR-125b

1

Acute myeloid leukemia

2

miR106a-363

2

Acute Promyelocytic Leukemia

3

miR-106b/25

3

Adenocarcinoma

4

miR1-1/133a-2

4

B-cell lymphomas

5

miR-130b~301b

5

Bladder cancer

6

miR-132/212

6

B-lymphoma

7

miR-134/487b/655

7

Breast cancer

8

miR-143/145

8

Cervical cancer

9

miR-144/451

9

Colon cancer

10

miR-15a/16–1

10

Colorectal cancer

11

miR-17-92a

11

Colorectal adenoma

12

miR-181a/b

12

Cutaneous squamous cell carcinoma

13

miR-183/182/96

13

Epithelial ovarian cancer

14

miR-191/425

14

Esophageal cancer

15

miR-193b/365a

15

Esophageal cell carcinoma

16

miR-193a/316

16

Esophageal squamous cell carcinoma

17

miR-194/215

17

Gastric Cancer

18

miR-199a/−214

18

Gastrointestinal cancer

19

miR-200b-429

19

Glioblastoma

20

miR-200c/141

20

Glioblastoma multiforme

21

miR-216a/217

21

Glioma

22

miR-221/222

22

Hepatocellular carcinoma

23

miR-224/452

23

Cholangiocarcinoma

24

miR-23a/24

24

Chronic lymphocytic leukemia

25

miR-23b/27b/24-1

25

Insulinoma

26

miR-29a/b1

26

Intestinal tumor

27

miR-302/367

27

Liver cancer

28

miR-371/373

28

Lung adenocarcinoma

29

miR-379/656

29

Lung cancer

30

miR-513b/513c

30

Malignant pleural mesothelioma

31

miR-424/503

31

Mantle cell lymphoma

32

miR-508/513a

32

Medulloblastoma

33

miR-509/514b

33

Melanoma

34

miR-512-1/519a-2

34

Multiple myeloma

35

miR514a-3/510

35

Neuroblastoma

36

miR-545/374a

36

Nonsmall cell lung cancer

37

miR-891b/892c

37

Oligodendroglioma

38

miR-99b/let-7e/miR-125a

38

Ovarian Cancer

42

Pancreatic Cancer

43

Parathyroid carcinomas

44

Prostate Cancer

45

Rectal cancer

46

Renal cell carcinoma

47

Retinoblastoma

48

T-cell leukemia

49

Testicular germ cell tumors

50

Thymoma

I am aware that preparing such a big work it is not easy in a short and direct way. However, authors should consider the utility of the work for the scientific community and the possibility for readers to extrapolate significant messages after reading.

The main message of the MS is that information about transcriptional regulation of miRNA clusters has much higher potential to be used in translational research than has occurred recently. This assumption is supported by the listing of several clusters that share the same basic properties, are down-regulated and show oncostatic capacity. Down-regulated clusters with these features can become promising multi-targets for therapeutic manipulation.

In addition to the abovementioned aim, we also intended to provide complex and exact information about work done in colorectal cancer research supported by experimental evidence. In this way, we provide an easy-to-use key to find positive or negative controls in experiments or to supplement planned research on particular miRNAs, lncRNAs or host genes. We believe that this increases the capacity of this MS to facilitate research in this field.

The first section of this statement is emphasized in the last sentence of abstract and conclusions. The second section is guaranteed by presence of supplementary tables 1-3 and GO analysis in supplementary file 4.

Main comments:

Lines 67-69: please modify the sentence in the most updated and correct way. The “second miRNA strand” is called passenger strand and the use of asterisk is obsolete (I think it has been changed already 2-3 miRBase versions ago). Please change with the designation of -3p and -5p arms.

- thank you for the comment, the description of the second strand was changed accordingly and mentioning of asterisk was omitted from text, please see tracked version of MS

Revise the sentence at lines 49-50. The function of Dicer is misleading here and not clear.

- thank you for the comment, sentence was changed to make statement more clear, please see tracked version of MS

I found that manuscript is not well balanced in the distribution of the topics. There are almost 2 pages dedicated to function and maturation of miRNAs (which is a bit out of topic) and then regarding lncRNAs (more connected with the manuscript), authors dedicated a mini and very reductive paragraph (lines 87-93). Moreover, the conclusive part (4. Conclusions) it is almost 2 pages long, which I found excessive for a “conclusion”.

- the first part of MS was shortened, however, we need to keep this part of MS at least in this shortened version since main text is focused on expression of transcriptional unit (cluster), therefore biogenesis description should be described.

- chapter “conclusions” was split into three parts to make text better balanced:

  1. Regulation of expression of identified clusters
  2. Target genes and functions of identified clusters
  3. Conclusions

please, see tracked version of text.

Regarding Conclusions section again, I suggest to authors to be more precise on the enrichment analyses performed.. The graphs relative to figure 1 from what type of enrichment analyses was done?? What is the program used? What is the list of genes used? The data are not even properly discussed and connected with the data reported for clusters.

- thank you for the comment. We agree that enrichment analysis deserve detailed description and therefore we provided it. However, it was overlooked.

Supporting data to figure 1, list of genes, information about the program, and even list of gene belonging to particular category was (and is) provided in Supplementary file 4. We also referred to references acknowledging The PANTHER Classification System as it was requested by their authors:

  1. Thomas, P. D.; Campbell, M. J.; Kejariwal, A.; Mi, H.; Karlak, B.; Daverman, R.; Diemer, K.; Muruganujan, A.; Narechania, A. PANTHER: a library of protein families and subfamilies indexed by function. Genome Res. 2003, 13, 2129-2141. Supplementary Materials. https://doi.org/10.1101/gr.772403
  2. Thomas, P. D.; Kejariwal, A.; Guo, N.; Mi, H.; Campbell, M. J.; Muruganujan, A.; Lazareva-Ulitsky, B. 2006. Applications for protein sequence-function evolution data: mRNA/protein expression analysis and coding SNP scoring tools. Nucl. Acids Res. 2006, 34, W645-W650. https://doi.org/10.1093/nar/gkl229

We added GO analysis outcome into chapter 5, please see tracked text.

I suggest a complete revision also of the initial part of chapter 2 (lines 95-102). The whole part does not make much sense to me.. “MiRNA measurement” is an improper term as well as “miRNA evaluation”. What the authors meant??Maybe they meant measurement of levels of expression? And miRNA extraction and methodology of measurement? Please, clarify it! And what does this sentence (“however, this inconsistency precludes these miRNAs from use as effective bio-tools”) mean ?

- thank you for the comment, text was corrected, please, see tracked version of MS.

- sentence “The reason for this may be due to differences in biological context or methods of miRNA evaluation; however,” was omitted from text.

Please revise the text because the authors declared on lines 100-102 that they described only clusters downregulated in CRC. This concept has been repeated for all the clusters subsections and in the last part of the manuscript (I have counted at least 15 times!!). This will help to make the manuscript lighter to read.

- thank you for the comment, we omitted this phrase in 8 occasions to make text lighter to read. Please, see tracked version of MS.

I think authors should try to connect better the evidence reported for each clusters with the fact that those clusters (or better, the miRNAs in the clusters) were reported frequently downregulated in CRC. This will be the additional information for readers that will make the text something more than just a shopping list.

- evidence reporting frequently down-regulated miRNA in clusters are provided in detail in supplementary table 2 that was originally part of MS but reviewer 2 suggested to remove it completely. We cannot ignore correction requested by reviewer 2 but we agree that reader should be able to access this information if he/she is interested in it. Therefore Table 2 was implemented into supplementary files. Information about decreased expression of identified clusters is available in many parts of MS (so many that we had to reduce them, as it was requested by reviewer 1). We hope that this solution is acceptable for both reviewers.

Minor:

-Line 84: change “into mature miRNAs” to “into a final mature miRNAs”

- thank you for the comment, a sentence was changed accordingly

-lines 178-188: I think this part is relative to the Table but it looks like part of the text (and redundant to page 4)

- lines 178-188 are really legend of the table; as it is very inconvenient to look for terms used in table in article text. We believe that informative legend significantly increases readability of the text. However, we omitted to the first sentence of the legend that is repeated on the page 4. Please, see tracked version of MS.
